# Recent Advances in Bioimage Analysis Methods for Detecting Skeletal Deformities in Biomedical and Aquaculture Fish Species

**DOI:** 10.3390/biom13121797

**Published:** 2023-12-14

**Authors:** Navdeep Kumar, Raphaël Marée, Pierre Geurts, Marc Muller

**Affiliations:** 1Department of Computer Science and Electrical Engineering, Montefiore Institute, University of Liège, 4000 Liège, Belgium; raphael.maree@uliege.be (R.M.); p.geurts@uliege.be (P.G.); 2Laboratory for Organogenesis and Regeneration (LOR), GIGA Institute, University of Liège, 4000 Liège, Belgium; m.muller@uliege.be

**Keywords:** fish, skeletal deformities, bioimage analysis, aquaculture, biomedical, image processing, computer vision, zebrafish, medaka, seabream

## Abstract

Detecting skeletal or bone-related deformities in model and aquaculture fish is vital for numerous biomedical studies. In biomedical research, model fish with bone-related disorders are potential indicators of various chemically induced toxins in their environment or poor dietary conditions. In aquaculture, skeletal deformities are affecting fish health, and economic losses are incurred by fish farmers. This survey paper focuses on showcasing the cutting-edge image analysis tools and techniques based on artificial intelligence that are currently applied in the analysis of bone-related deformities in aquaculture and model fish. These methods and tools play a significant role in improving research by automating various aspects of the analysis. This paper also sheds light on some of the hurdles faced when dealing with high-content bioimages and explores potential solutions to overcome these challenges.

## 1. Introduction

In the realm of biomedical research, model fish species like zebrafish (*Danio rerio*), and medaka (*Oryzias latipes*) are highly regarded as valuable vertebrate models. They are extensively used in a variety of biomedical applications, encompassing drug testing, morphometric screening, genome editing, toxicology assessments, and behavior analysis in vertebrates [1,2,3,4,5,6,7,8]. These model fish exhibit significant genetic and metabolic pathway similarities to both fish and mammals, sharing over 70% of their genes with humans [9,10,11,12]. Notably, zebrafish and medaka models are particularly advantageous due to their ease of maintenance and reproduction. Together with other technical advantages such as small size, low maintenance cost, high fecundity, and amenability to genetic engineering tools, the reason these fish are so popular among scientists is their suitability for in vivo imaging [13,14]. The embryonic and larval stages of these animals are translucent, allowing for the application of advanced imaging technologies to observe biological processes in a living animal. This property bears great potential for biomedical research when combined with the availability of transgenic and mutant lines that allow modeling human skeletal diseases and tracking specific organs and cell types with fluorescent markers [15]. Such characteristics not only offer an incredible tool for fundamental research, but also greatly benefit drug discovery. According to the Business Research Insights website, the global zebrafish model services market size was USD 434.4 million in the year 2022 and is projected to reach USD 618.23 million by the year 2031, with a compound annual growth rate (CAGR) of 14.4% during the forecast period.

Fish is recognized as a valuable source of high-quality protein and essential nutrients that are integral to a healthy human diet. Within the aquaculture industry, fish holds a primary position as the predominant source of cultivated seafood for human consumption. According to the European Commission’s Ocean and Fisheries website, marine and freshwater fish constitute approximately 49% of the total aquaculture production. Commonly consumed food fish species include gilthead seabream (*Sparus aurata*), meagre (*Argyrosomus regius*), and salmon (*Salmo salar*), which are saltwater species, while rainbow trout (*Oncorhynchus mykiss*) is a freshwater counterpart. In their natural habitats, such as the sea or rivers, healthy fish thrive without external interventions in terms of food and care. However, in fish farms, fish are reared within controlled or artificial environments, such as ponds, tanks, or cages, which necessitate external care and provisioning of food. Given the escalating global demand for aquaculture products, the industry faces significant pressure to enhance its supply. To meet this demand, fish farmers adopt intensive production practices, which can result in challenges like deteriorating water quality, higher fish density per unit of water volume, and limited food availability for the fish. These factors may contribute to stressed fish, the development of physical abnormalities, and susceptibility to serious diseases [16]. Fish with disease or deformities are rejected by the potential retailers or customers, thereby representing a significant economic loss to the fish farmers [17]. Major economic losses are directly due to the development of skeletal disorders altering the external shape of reared fish, i.e., opercular and vertebral column deformities [18]. Moreover, tedious technical effort and time are required to manually cull out the deformed fish from the productive cycle, which should be carried out as early as possible in order to not waste resources on growing suboptimal fish.

To detect and classify the deformities in the reared or model fish, manual inspection or analysis is employed, which requires significant time and technical effort. Moreover, direct physical interaction with the fish can induce fear or stress that may reflect on its behavior. Due to abnormal behavior or stress, fish can not swim or take proper diet, which can lead to poor health of the fish [16]. To improve animal welfare both in aquaculture and biomedical research, scientists are looking for methods requiring minimal manual interaction with the animals, with more focus on their health and quality of life. Computer vision is one such area that is increasingly being adopted by fish farmers and biomedical researchers to monitor the health and/or behavioral changes of the animal/fish. It may be helpful in identifying the causes of fish stress or any health hazard with minimal interaction with the animal. Computer vision and image processing techniques can also be helpful to speed up other routine procedures such as animal feeding [19], animal sorting, and animal counting by automatizing these tasks. According to the website, https://this.fish/blog/ai-guide-tracking-ais-explosive-growth-in-aquaculture/ (accessed on 2 December 2023) *“this.fish”*, the top 10 artificial intelligence (AI) and software start-up companies for the aquaculture industry have raised USD 282 million in the past 5 years, illustrating the importance and prevalence of AI-based smart farming in aquaculture.

Nowadays, automatic or semi-automatic computer-vision-based image processing techniques are being used in aquaculture industries and biomedical research to speed up the detection and diagnosis of diseases in the fish under study. These computer-vision-based techniques employ artificial intelligence methods such as machine learning or deep learning, which not only speed up the diagnosis but are also helpful in improving the accuracy of the detection. Deep learning represents a cutting-edge AI approach that empowers computers to learn from data and perform tasks on par with human capabilities. It utilizes multi-layered neural networks to directly acquire task-specific features from the data and make informed decisions when confronted with unseen data after the learning process. This methodology avoids the use of manually engineered features (as used in older image processing approaches) and, instead, defines a learning process that autonomously extracts features from the data, reducing the need for human intervention. In scenarios involving image data, such as image analysis, specialized types of neural networks known as deep convolutional neural networks (CNNs) have been designed. Such architectures play a pivotal role in computer vision applications, including tasks like object classification, object recognition, segmentation, and object counting. For the analysis of biomedical images, deep-learning-based convolutional neural networks (CNNs) are widely employed and increasingly favored for their accuracy, which rivals human performance [20]. Although debatable, CNNs are considered an imitation of the working of neurons in the human brain for visual perception and understanding objects in the images. They use a set of consecutive convolutional blocks or layers in order to understand the useful patterns for recognizing the objects in the images. The fundamental component of a CNN is the *convolutional layer*, comprising a collection of filters (or kernels), whose parameters are trained during the learning process [21,22]. Convolutional layers excel in extracting features from images by addressing spatial redundancy through weight sharing, and the features become more distinct and informative while going deeper into the layers. The role of the *activation layer* is to fire or activate the particular neurons while processing the information, and spatial invariance is achieved using *pooling layers*. In the end, a condensed feature representation is generated, encapsulating the essential content of the image in *fully connected layers* [23]. A typical CNN architecture for object classification tasks in images is shown in Figure 1.

In this paper, we review various computer-vision-based automatic and semi-automatic image analysis methods and tools that are used in morphometric and phenotype studies of the aquaculture and biomedical model fish. Our literature survey was performed using searches in PubMed, Scopus, Google Scholar, Web of Science, Bioimage Informatics Index (https://biii.eu (accessed on 2 October 2023)), and Papers With Code (https://paperswithcode.com/ (accessed on 5 October 2023)) databases, and thanks to personal communications with researchers in the field, including members of the BioMedAqu project. A related, recent review appears in [24]. The latter, however, primarily focuses on the application of AI algorithms for behavioral analysis, genomics and neuroscience in zebrafish research, while our paper concentrates on the application of AI algorithms in the analysis of bone-related deformities in biomedical and aquaculture fish species.

Our paper is structured as follows. In Section 2, we first highlight the popular imaging techniques used in acquiring fish bioimages. In the following sections, we delve into the image analysis techniques employed in biomedical (Section 3) and aquaculture investigations (Section 4) to identify and categorize different types of bone-related deformities in both model and food fish species. Table A1 in Appendix A focuses on user-friendly, AI-based image analysis tools used in fish morphometric and phenotype research along with their specifications. Finally, we also highlight the challenges encountered during the image analysis tasks (Section 5) due to image acquisition settings, data scarcity, and algorithmic constraints and requirements.

## 2. Imaging Techniques Used in Fish Bioimages

One of the main advantages of using zebrafish as a model animal over other animals is its transparent body during early, external development life stages, especially from 0 to 10 days post-fertilization (dpf). The transparent body of the larva makes it easy for the biologists to see through its developing organs and bones during in vivo studies and also helps to produce bioimage datasets using various image acquisition equipment [25,26]. Given that image acquisition precedes image analysis, it is crucial to employ suitable imaging methods and protocols to ensure effective and accurate image analysis, particularly when conducting AI-based image analysis. Due to the small size of the zebrafish and medaka embryos and larvae, advanced optical microscopy imaging methods are employed to capture maximum information at the microscopic level [27]. Microscopic imaging methods necessitate a meticulous pipeline to be adhered to, ensuring the prevention of unwarranted variations in acquisition adjustments and parameters that might introduce artifacts capable of influencing the outcomes of image analysis algorithms [28]. Beyond fundamental considerations like luminosity and focus control, special attention to the fish’s positioning and the characteristics of the glass plates is also needed to mitigate potential issues related to light refraction. This precautionary approach aims to prevent problems like shadowed areas in the images that could disrupt the subsequent analysis [29]. Since most phenotype and morphometric studies in biomedical research require capturing the fine-grained information at the sub-cellular level, microscopy methods such as bright-field or fluorescence microscopy are prevalent compared to other imaging approaches [30,31]. More recently, confocal and light-sheet microscopy deliver three-dimensional images [32], while Raman spectroscopy, Fourier-transform infrared spectroscopy, or mass spectrometry imaging are able to reveal the spatial distribution of individual (bio)molecules or classes of molecules [33,34,35,36], resulting in ever more high-content and demanding analysis requirements.

Apart from microscopy methods, X-ray radiography techniques are also popular in biomedical and aquaculture research for analyzing the skeletal structures of the juvenile and adult fish, including microCT imaging [37,38,39]. While microscopy imaging methods are employed in the early life stages (embryonic and larval) of the model fish due to its body’s optical clarity and small size, radiography methods are employed in the later life stages to visualize hard tissues. The adult model fish serves as a distinct and valuable resource for studying pathogenic and therapeutic aspects of adult human bone diseases. This is attributed to the fact that certain functions such as bone turnover, repair, degeneration, and metabolic responses are not fully mature in embryos [40]. Similarly, in aquaculture research, radiography imaging methods are utilized for juvenile and adult fish for several types of phenotype and morphometric studies [41,42].

## 3. Bioimage Analysis in Biomedical Research

In recent years, considerable progress has been obtained in the field of AI development. In particular, deep-learning techniques are used for automatic image analysis in biomedical sciences and are becoming the predominant choice in various morphometric and phenotypic studies [43,44]. For single-cell phenotype assays that require gathering intricate data at the cellular or sub-cellular level to discern features linked to cellular shape, protein localization, and intracellular movement, classifying phenotypes poses a notable challenge. This challenge becomes more critical in scenarios involving high-content screening where many high-resolution images are acquired. In such situations, deep-learning methods have proven to be more effective than conventional approaches, as evidenced by the research of [45,46]. In morphometric and phenotype research, segmenting various parts of the fish is a key operation for analyzing development in the fish. For example, EmbryoNet (version 1.0.1) [47] is a deep-learning-based software method to identify the phenotype defects in the embryonic stage of the zebrafish. The aim of this approach is to bridge the gap between observed phenotypic traits in embryos and the underlying molecular signaling pathways responsible for those traits. The diverse set of observable traits or phenotypes that researchers monitor during embryonic development include morphological changes, cellular behaviors, or other features. The problem of phenotype classification in zebrafish in high-throughput screening using the end-to-end deep-learning method is described in [48]. In this study, the authors tackle the challenge of categorizing morphological alterations in zebrafish found in multi-fish wells, which often have fish overlapping with one another. Many morphometric studies involve analyzing the skeletal parts of the fish, especially the jaws, operculum, and vertebral column. In order to analyze these parts, researchers in the field of skeletal biology largely rely on advanced imaging techniques followed by morphometric analysis of skeletal structures in animal models and human patients [49]. This makes it possible to explore how specific molecular mechanisms translate into phenotypic changes in both physiological and disease contexts. The great advantage of such an approach is that it allows for translating qualitative biological information into mathematical language, amenable to statistical analysis. Analyzing malformations during early developmental stages offers other valuable insights into the potential toxicity of chemicals. In [50], an automated software tool is developed to quantify the morphometric defects during developmental toxicity screening in zebrafish embryos. In this approach, morphometric features are extracted and organized in a hierarchical manner using length and surface areas from contour information of different parts of the zebrafish candidate. In order to detect certain features, information about previously detected features should also be included. Finally, the detected features are the boundary coordinates of the contours of the objects such as eyes, head, and swim bladder. of embryos. One of the highly sensitive indicators of developmental toxicity is the absence or presence of abnormalities in the tail, spine, or other parts of the fish.

In the work of [29], a machine-learning-based phenotype classification method was developed in which the pipeline comprises a dual-step strategy, beginning with a three-class classification model. Initially, it separates images of “Dead” or “Chorion” from “Normal” embryos before addressing other anomalies such as “downward curved tail”, “upward curved tail” or “short tail” etc. When contrasted with manual classification, the automated classification using machine learning yields a consensus agreement with biological experts’ voting in nine out of eleven assessed defects in 3-day-old zebrafish larvae, ranging from 90% to 100%. Another supervised machine learning approach to automatically classify the absence or presence of malformation in the spine of the medaka fish embryo is discussed in [51]. In that work, a dataset of 2D high-resolution microscopic images of medaka fish is used. Feature extraction is performed, first by segmenting the embryo from the images. Since many malformations are characterized by abnormal spine curvature, features such as body length or curvature angle are then extracted. These features are then fed into a machine-learning-based classifier for training. Since feature characterization depends on the geometry of the body representation of the embryo, the authors admitted that their method is not applicable when the tail of the deformed embryo makes a hook shape, hence not universal to any type of malformation or failure with a high degree of severity in the deformed tail. Figure 2 shows different types of deformities in the tail part of the zebrafish embryo.

Many morphometric studies involve analyzing the skeletal parts of the fish, especially jaws, operculum and the vertebral column. In order to analyze these parts, researchers in the field of skeletal biology largely rely on advanced imaging techniques followed by morphometric analysis of skeletal structures in animal models and human patients [49]. This makes it possible to explore how specific molecular mechanisms translate into phenotypic changes in both physiological and disease contexts. The great advantage of such an approach is that it allows for translating qualitative biological information into mathematical language, amenable to statistical analysis. In the past few years, several zebrafish in vivo assays for evaluating the osteogenic and mineralogenic activity of natural compounds and synthetic drugs were developed [13]. Among these, the morphometric analysis of the opercular bone in the zebrafish larva allows for an easy, cost-effective and fast evaluation of the osteogenic bioactivity of molecules of interest [52]. As pointed out in [53], analyzing the areas of the head and operculum regions in zebrafish larvae, along with quantifying the operculum-to-head ratio, serves as a reliable indicator of altered bone formation and mineralization. This method is well established and validated for screening bioactive molecules with potential effects on bone health. As such, this assay has been successfully implemented to identify natural extracts with promising pharmaceutical potential. However, obtaining biological quantitative data from microscopic images is still largely performed manually. With the aid of image processing software, operators manually segment specific regions of interest (e.g., the eye, the head, or the trunk), and extract data such as pixel areas or fluorescence intensities [53]. This laborious and time-consuming process also suffers from poor reproducibility due to operator-introduced biases. A set of various tools for the popular open-source image analysis software such as *ImageJ* (v1.54g) were recently developed to increase the automation of image processing obtained from the zebrafish operculum and other zebrafish-based biological assays [54]. However, such tools still require substantial input and supervision from the operator [41,55,56].

In [57], a deep-learning-based image segmentation method is proposed for the segmentation of the head and operculum area of zebrafish larvae from red channel fluorescence microscopy images. This approach allows for the complete automation of operculum and head area segmentation, greatly reducing the variability related to operator-introduced biases and, importantly, the time needed for extracting biological data from microscopic images, therefore substantially increasing the throughput of such an assay. This method reports a “Dice score” of 95%, which is calculated as the ratio of the overlapping area between the actual and predicted regions, excluding the background. The model (https://github.com/navdeepkaushish/ (accessed on 4 November 2023)) is integrated in an open-source, web based image analysis tool called Cytomine [58]. In Figure 3, deep-learning-based operculum and head segmentation is shown.

At later stages, skeletal development in teleosts progresses under the influence of various biotic and abiotic factors. Consequently, the skeletal characteristics of these organisms are constantly shaped by environmental elements that interact with their genetic makeup. Rearing density has been described as an environmental variable that influences skeletal development, especially vertebral column formation in both small model and larger aquaculture fish, and high densities have been reported as driving factors that induce skeletal anomalies. In the work of [56], the authors provide comprehensive insights into the postcranial components of the medaka skeleton, and they investigate the structural and cellular transformations associated with the emergence of skeletal anomalies. They give an assessment of how rearing density impacts specific meristic counts (e.g., number of vertebrae) and the variability in the occurrence and types of skeletal irregularities. A reliable skeletal reference is provided to establish optimal laboratory conditions that can be used for the assessment of future experimental setups. The work described in [59] implemented a deep-learning model to automatically locate six important landmark positions on the vertebral column of the medaka fish, thereby assisting the biomedical researchers in speeding up the image analysis. The landmarks on the vertebral column are able to detect either upwards/downwards shifts or deviations from the normal axis or elongations/restrictions of the regions which could indicate vertebral body fusions or extranumerary vertebral bodies. Figure 4 shows six landmark points on the microscopy image of a medaka juvenile (40 dpf). For deep-learning model training and evaluation, a total number of 430 Alizarin-red-stained microscopy images of 2560×1920 size were used. In terms of performance, the authors reported an average Euclidean distance of about 9 pixels between actual and predicted landmark locations, estimated on a set of images that were not seen during model training. The model is integrated into the Cytomine image analysis tool [58].

Osteoporosis is a metabolic bone condition resulting from an imbalance in the processes of bone formation and resorption. This disrupted equilibrium leads to weakened bones that are prone to fractures, even from minor incidents. With an increasingly aging population, the prevalence of osteoporosis is on the rise, and currently, there are limited pharmaceutical interventions available, particularly for enhancing the bone-building activity of osteoblasts [60]. Encouragingly, extensive genetic studies conducted on individuals with low (LBM) and high bone mass (HBM) and genome-wide association studies (GWAS) in the general population have revealed fresh insights into pathways containing key players in bone formation [61,62,63,64,65,66]. These discoveries hold the potential for the development of new drug targets. Such studies involve analyzing bone structures of the zebrafish model in its earlier development stages. For quantification of the size and shapes of bone structures, anatomical landmark points are assigned to them for the geometric morphometric analysis. Manually annotating these landmarks is a tedious and time-consuming process that requires technical expertise as well. The work described in [59] uses deep learning to automatically detect the landmark locations of various bone structures in zebrafish larvae, thereby making the task of morphometric analysis easier for biologists. In this work, 25 landmark locations for bone structures are targeted (see Figure 5 for a sample image and its annotations), and a total of 113 microscopy images with size 2576×1932 are used for the training and the evaluation of several CNN-based deep-learning models. The authors reported an average distance between predicted and actual landmark locations of about 11 pixels. The trained model is integrated into the Cytomine image analysis tool as a software package called “S_Deep-Fish-Landmark-Prediction” (v1.2.0).

## 4. Bioimage Analysis in Aquaculture

The aquaculture industry plays a crucial role in providing a substantial food source for human consumption [67]. However, it faces mounting pressure to meet the growing demand for food supply. A decade ago, farmers relied on manual methods for various tasks, including sorting fish by size, identifying diseased individuals, removing deformed or deceased fish from healthy stock, and counting fish numbers. In addition, traditional invasive methods involve physically removing fish from the water and having an expert inspect them to identify potential diseases [68]. In recent times, deep-learning-based image processing techniques have emerged as an approach in the aquaculture sector to address the limitations of manual and labor-intensive procedures [43,69]. These methods offer the advantage of being less time-consuming and requiring less technical expertise. Importantly, they are non-invasive by nature, contributing to improved fish health and the promotion of sustainable aquaculture practices.

Although traditional machine learning techniques have been employed in aquaculture research for over a decade, there is a recent surge in the popularity of computer vision-based deep-learning methods, primarily because of their high performance and easy-to-use traits [70]. While relatively new in the aquaculture field, deep-learning analysis methods are proving beneficial by expediting routine tasks for fish farmers in a non-intrusive manner [69]. Simultaneously, they assist technicians and researchers within the aquaculture industry in identifying and categorizing fish disorders and deformities.

Recently [71], a method based on deep learning has been introduced for the non-invasive measurement of meagre fish length and weight. The approach uses stereo images of fish as inputs of a deep-learning object detector. This detector identifies and outlines individual fish by creating bounding boxes around them. Subsequently, each bounding box is employed to extract an image of the individual fish, which is then processed through a pre-trained CNN to pinpoint two significant landmarks on the fish: the snout tip and the base of the middle caudal rays. A landmark detection algorithm calculates the fish’s length in pixels by measuring the distance between these two landmarks. Finally, the pixel-based length is converted to physical units by using calibration data.

The European aquaculture sector places increasing importance on the production of gilthead seabream (*Sparus aurata*) [17]. However, the occurrence of skeletal deformities in farmed gilthead seabream poses a significant challenge for the industry [72]. This issue results in economic losses, negatively affects consumers’ perception of aquaculture, and raises concerns about the welfare of the fish [73]. While previous efforts have predominantly concentrated on reducing the occurrence of skeletal anomalies during the hatchery phase, current research is directed toward addressing the subsequent pre-ongrowing phase, where more severe deformities impacting the fish’s external appearance often occur [41]. Considering that the pre-ongrowing phase primarily occurs in land-based facilities, it serves as the final opportunity for farmers to implement quality controls and remove deformed fish before transferring them to sea cages. Additionally, the pre-ongrowing phase offers a more effective means of sorting and quality assurance compared to culling during or at the end of the hatchery phase. While fish with severe cranial anomalies can be readily identified and removed at the end of larval rearing due to the early development of their skull bones, vertebral axis deformities are infrequent at this stage, and they often manifest in more advanced developmental stages, which makes them harder to detect [74]. The work in [41] aims to ascertain whether tank volume or stocking density is a primary factor influencing the development of skeletal anomalies during the pre-ongrowing phase in gilthead seabream. The authors found that higher stocking densities led to a higher incidence of specific cranial and axial deformities in the fish. This research provides valuable insights for the aquaculture industry and offers practical applications for fish farmers to enhance the skeletal and morphological quality of farmed gilthead seabream. The research is further exploited with the following observations in mind: (1) Bone elements that were incompletely fused were treated as separate elements; (2) Supernumerary bones with typical morphology were not classified as anomalies but were accounted for in variations in meristic counts. Conversely, supernumerary elements with abnormal shapes were categorized as anomalies; (3) Only variations in shape that were distinctly and unequivocally identifiable were recognized as skeletal anomalies. For instance, deviations in the vertebral axis that were linked to deformation of the involved vertebrae were considered anomalies. Such morphometric features are marked by selecting 19 landmark locations over the fish. Any deviations in these landmark points indicate a potential anomaly in the vertebral column and overall shape of the fish. A sample image from the dataset with annotated landmarks is shown in Figure 6.

The work of automatically identifying these landmark locations is described in [59], where a deep-learning methodology was devised to identify the landmark locations on the radiography images of gilthead seabream. The deep-learning-based CNN is trained on 748 images of varying sizes and is evaluated on 100 test images, yielding an average distance of about 5 pixels between true and predicted landmarks. The method is integrated into the Cytomine image analysis tool.

## 5. Challenges in Bioimage Analysis Tasks

While traditional and deep-learning methods are valuable for various tasks in biomedical and aquaculture research, they are often challenged by complexities arising from factors like image acquisition, noise, multi-modality, or domain changes. Many of the methods described earlier are tailored to specific issues or rely on particular fish species and their image characteristics. A universal model capable of simultaneously addressing multiple tasks does not currently exist. In this section, we underscore the challenges posed by these factors and their implications for image analysis tasks. We also explore potential solutions that can be employed to tackle these issues.

### 5.1. High-Content, High-Throughput Imaging

Modern imaging instruments, combined with robotic devices, lead to vast amounts of high-resolution images that allow more specific and detailed analysis [5,75,76]. However, they will also generate terabytes of data with gigapixel images. Moreover, different types of confocal microscopy, time-lapse (video), or microCT methods deliver three-dimensional images that further increase the data load [31,33,34,35]. Analyzing and handling high-content images and observing features at the cellular or sub-cellular level become a bottleneck. Even for the current state-of-the-art analysis methods, it is very challenging to find informative patterns for processing vast amounts of complex structured data. The challenges encountered while processing these high-content image data of zebrafish have previously been discussed in [27] with regard to image quality, annotation, and storage. With the huge size of high-content images, it becomes imperative for the current existing image processing methods to process them in more standardized and generic ways. The authors emphasized the general requirement for the design of a workflow, information on the imaging data (metadata information) being analyzed, and availability of the storage infrastructure such as data center support. Image processing methods must include some preprocessing for noise reduction, correction of inhomogeneous illumination, and correction of attenuation before applying analysis methods. Usually, these preprocessing steps are application independent and can be easily found in many standard image processing toolboxes. While deep-learning-based approaches are better performing than their traditional counterparts in image analysis tasks, they still suffer from performance issues while dealing with high-resolution biomedical imaging. If the objects are very small in size, have distorted shapes, or the annotations are not fully outlined, these models can not perform well. Proper and careful preprocessing might be needed before being fed into neural networks. Another issue while using deep-learning-based models is caused by the fact that images have to be resized (usually downsized and tiled) to make them suitable for use in CNNs. As microscopic or histological images are of very high resolution, downscaling or squeezing them will lead to extensive loss of pixel-level information that should be retained during the study of the micro-anatomy of cells, tissues, and organs. Strategies such as multiple instance learning and weakly-supervised learning are currently being developed to deal with very large images in digital histology [77] and might be similarly useful in high-resolution fish imaging.

### 5.2. Choice of Image Analysis Methods/Protocols

Traditional image analysis methods that use simple image processing functions are good when the information they process is small. With the growing dimensionality of the data, the use of traditional methods is time-consuming and requires a lot of human intervention with technical expertise to analyze the data. Artificial intelligence (AI)-based techniques, especially deep-learning methods, are helpful to speed up the analysis by making the process semi- or fully automatic. However, training deep-learning-based CNN for high-content image analysis is an uphill task, and applying them effectively in biomedical image analysis tasks is still challenging. In most biomedical images, preserving spatial information is very crucial; thus, if the architecture of the CNN does not preserve the spatial information during training, it may fail to produce the desired results. The choice of the neural network architecture is also dependent upon the imaging data being used for training. Another prominent challenge to deal with in biomedical image analyses is a class imbalance, occurring, for example, when segmenting small cells or tissues of a few pixels areas from large-sized images [57,78,79]. Proper care is needed to take this problem into account while designing an image processing pipeline. Overall, there is also a need for more standardized implementations of image processing workflows (including software dependencies, input and output data formats, and application programming interfaces) to enable reproducible benchmarking [80] and to ease the choice of the best-performing method on a given problem. In addition, tools to foster interdisciplinary collaboration between biologists and computer scientists, providing easy ways to annotate images, execute, and proofread algorithm predictions, are also required [81].

### 5.3. Lack of Annotated Bioimages

The scarcity of well-annotated bioimage data is one of the prime concerns while designing an image analysis tool or method for biomedical research that requires annotated datasets to be trained, tuned, and validated. Natural image data sets are available in abundance on different platforms with open-source access. In contrast, bioimage datasets are acquired with expensive instruments in a laboratory in controlled conditions and are not easily accessible to the common people. The majority of bioimage datasets originate from either patients or model animals, and they are often restricted to specific laboratories or individuals. Furthermore, these datasets typically consist of a relatively small number of images, ranging from a few hundred to a few thousand, in contrast to natural image datasets that often encompass millions or even billions of well-annotated images. A limited set of well-annotated images is frequently inadequate for training deep-learning models, leading to suboptimal performance. Models trained with insufficient image data often struggle, making it challenging for analysts to deploy them effectively for image analysis tasks. One solution to address the problem of data availability is to use deep transfer learning approaches where a pre-trained and well-optimized CNN is used for training on small bioimage datasets. The authors in [82] exploit the deep transfer learning approaches to address the challenges faced in bioimaging due to its complex structure and lack of data availability. Another solution to deal with small datasets is to use data augmentation [83] techniques in which the data set is increased on the fly during training, using some image processing functions such as random rotation, shift, crop, shear, etc., depending upon the requirement. There are ongoing initiatives to set large repositories of fish bioimages with different modalities (microscopy, radiography, etc.) [84], and the European Union encourages researchers and academicians to provide their annotated bioimages for open-access research. This initiative further inspires fish communities to build substantially large fish bioimage datasets that would be helpful in advancing the research in aquaculture.

### 5.4. Miscellaneous

Although computer-vision-based deep-learning models outperform other traditional computer vision image processing methods and are increasingly becoming a favorite choice among the biomedical research community, they still suffer from a lack of interpretability. Even in general computer vision tasks, deep learning is mostly considered as *black box* learning, and research is still ongoing on how to interpret its results and learning mechanisms [85]. Moreover, medical images are content-sensitive, i.e., pixel-level information is needed, and resizing or squeezing to the size required by the network architectures adversely affects the performance due to loss of information. In aquaculture, real-time deployment of AI models for actually sorting the fish will require thorough testing of the working conditions (underwater, while swimming, in the air) and may be challenging due to variations in lighting, pose, background, occlusion, and water turbidity [86], leading to a decrease in the performance of the AI models. In such a complex setting, a considerable number of sensors, cameras, and other equipment must be deployed to gather and analyze real-time information, resulting in a substantial upfront capital investment [69]. Deploying large AI models is also demanding in terms of memory requirements, and incorporating these models into small memory chips poses a significant challenge. These challenges can be addressed using smart automation systems that include the integration of artificial intelligence, robotics, Internet of Things (IoT), edge computing, and advanced distributed algorithms for real-time image analysis [87,88,89]. Real-time applications of AI models in aquaculture would include, among others, fish counting, fish monitoring, fish feeding, fish classification (normal or deformed), fish behavior analysis, or fish sorting. From a biomedical imaging point of view, deep-learning-based models should be designed in more generic ways so that they can be trained without losing much of the information from the high-resolution images. Moreover, most biomedical researchers are from a non-mathematical background, and they might be more interested in knowing the objective interpretation of *why* rather than the subjective interpretation of *how* these deep-learning-based neural networks are predicting the discriminating features that often defy the common logic of human interpretation.

## 6. Conclusions

This paper highlights the current state-of-the-art machine and deep-learning methods being used in biomedical and aquaculture research for morphometric studies related to bone development. In this context, we describe the application of various deep-learning methodologies that can be applied in several bioimage analysis tasks, such as fish body part segmentation or anatomical landmark identification. The methods are not only helpful to speed up the image analysis, but also efficient in terms of accuracy, objectivity, and ease of availability. 

## Figures and Tables

**Figure 1 biomolecules-13-01797-f001:**
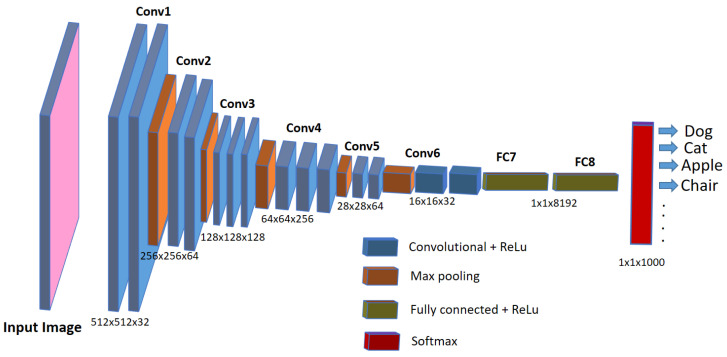
A typical CNN architecture for image classification tasks.

**Figure 2 biomolecules-13-01797-f002:**
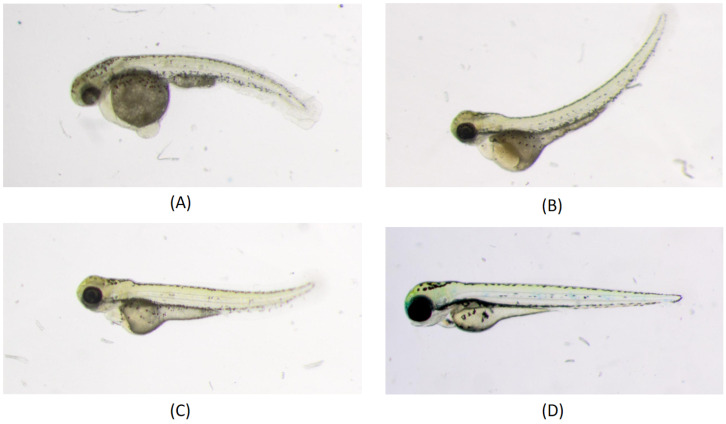
Different phenotypes in zebrafish tail. Larvae were imaged live under a dissecting microscope under transmitted light illumination: (**A**) Downward curved tail; (**B**) Upward curved tail; (**C**) Short tail; (**D**) Normal phenotype [29].

**Figure 3 biomolecules-13-01797-f003:**
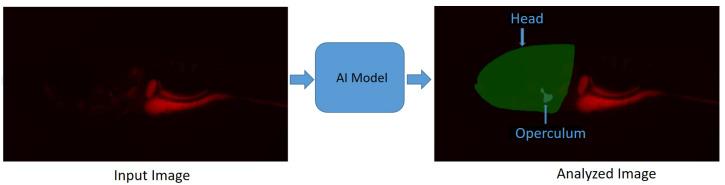
Segmentation results obtained with deep learning on a zebrafish image [57]. Zebrafish larvae were stained for mineralized tissues using Alizarin red and imaged using a fluorescence microscope: (**Left**) Input image; (**Right**) Analyzed image.

**Figure 4 biomolecules-13-01797-f004:**
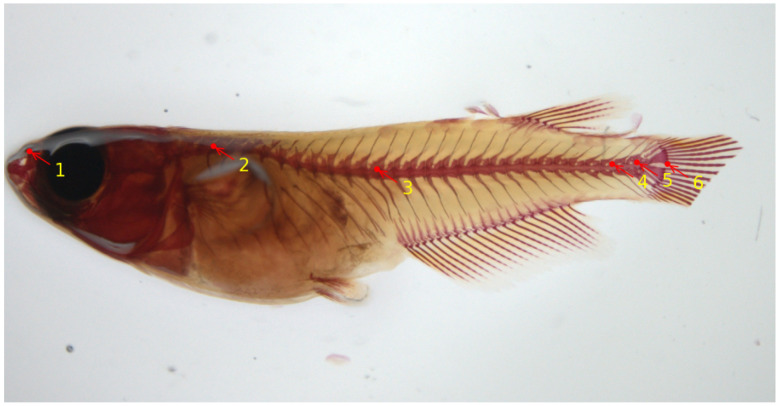
Six landmark locations on a medaka juvenile stained with Alizarin red for calcified tissues from microscopy dataset [59]. These landmarks are as follows: 1: rostral tip of the premaxilla; 2: base of the neural arch of the 1st (anteriormost) abdominal vertebra bearing a rib; 3: base of the neural post-zygapophyses of the first hemal vertebra (*viz.*, vertebra with hemal arch closed by a hemaspine); 4: base of the neural post-zygapophyses of the first preural vertebra; 5: base of the neural post-zygapophyses of the preural-2 vertebra; 6: posteriormost (caudad) ventral extremity of the hypural 1.

**Figure 5 biomolecules-13-01797-f005:**
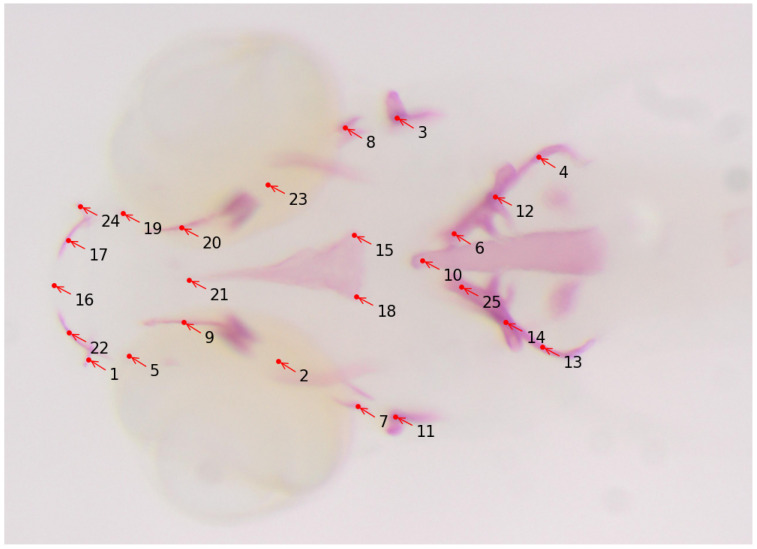
Sample image of zebrafish with 25 landmark locations. Alizarin red staining was performed on fixed larvae and imaged under a dissecting microscope. The landmark locations are annotated as follows: 1 and 24: Maxilla; 2 and 23: Branchiostegal ray 2; 3 and 11: Opercle; 4, 12, 13, and 14: Cleithrum; 5 and 19: Anguloarticular; 6 and 25: Ceratobranchial; 7 and 8: Hyomandibular; 9 and 20: Entopterygoid; 10: Notochord; 21, 15, and 18: Parasphenoid; 17 and 22: Dentary; 16: showing anterior end marking.

**Figure 6 biomolecules-13-01797-f006:**
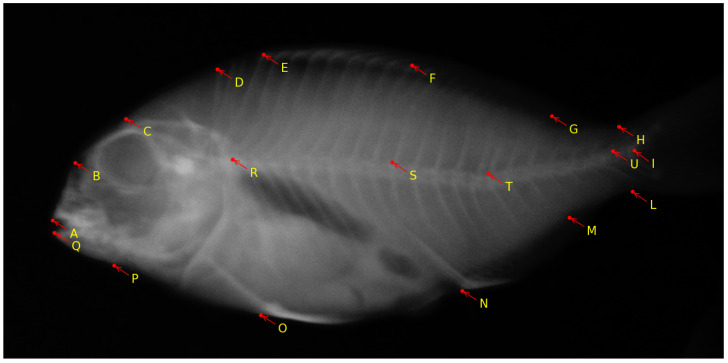
Sample X-ray image of pre-ongrowing phase gilthead seabream with 19 landmarks [59]. The image was obtained in air, on euthanized fish. The landmark locations are as follows: A: frontal tip of premaxillary; B: rostral head point in line with the eye center; C: dorsal head point in line with the eye center; D: dorsal extremity of the 1st predorsal bone; E: edge between the dorsal 1st hard ray pterygophore and hard ray; F: edge between the dorsal 1st soft ray pterygophore and soft ray; G: edge between the dorsal last soft ray pterygophore and soft ray; H: dorsal concave inflexion point of caudal peduncle; I: middle point between the bases of hypurals 2 and 3 (fork); L: ventral concave inflexion point of caudal peduncle; M: edge between the anal last pterygophore and ray; N: edge between the anal 1st ray pterygophore and ray; O: insertion of the pelvic fin on the body profile; P: preopercle ventral insertion on body profile; Q: frontal tip of dentary; R: neural arch insertion on the 1st abdominal vertebral body; S: neural arch insertion on the 1st hemal vertebral body; T: neural arch insertion on the 6th hemal vertebral body; U: between the pre- and post-zygapophyses of the 1st and 2nd caudal vertebral bodies.

## Data Availability

No new data were created or analyzed in this study. Data sharing is not applicable to this review article.

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
