# Peer review of "Recent Advances in Bioimage Analysis Methods for Detecting Skeletal Deformities in Biomedical and Aquaculture Fish Species"

_biomolecules, 2023, doi:10.3390/biom13121797_

Round 1
Reviewer 1 Report
Comments and Suggestions for Authors
This is a review paper on the latest image analysis technologies and development of AI tools for bone-related deformities in aquaculture and model fish. The important methods are introduced and reviewed, while the main challenges and potential solutions are also described. The topic of this review paper is interesting. It is an important problem to address. Therefore, this review paper can be potentially helpful for researchers in relevant domains. However, some critical questions need to be addressed first.
1. As a review paper, the total reference number (44 references in this case) is too low. Generally, for a full review paper, ~ 100 references would be expected, unless this is really a very small research area. The authors should do a more comprehensive literature search to cover more relevant works. Meanwhile, the authors should describe in the manuscript which database was used for literature search, what key words and key word combinations were used for the literature search, roughly how many papers were screened for this review paper.
2. What is the current practice of detecting such problems, only manual inspection? There should be existing industrial solutions to the problem. Please include an introduction to existing industrial solutions.
3. Please include the introduction of the imaging systems used for fish imaging. Is the imaging system inside the water or outside in the air? Are there any specific requirements of the spatial and temporal resolution of the imaging system? What are the typical wavelengths used for imaging?
4. Please include the basic structure of the AI model, instead of simply using ‘AI model’ in the figures. The authors should illustrate the basic structures like input layers, convolution layers and output layers. The authors can find relevant figures from the reference papers on AI.
5. What is the typical training data amount needed for such AI models?
6. In the challenge section, automated real-time imaging and analysis seem to be a big challenge. The authors should explain more about this challenge and potential solutions like robotics and automated computer vision / AI.
7. What is the estimated market size of such solutions? Is it an important industry problem to solve, or just an academic problem without much market value? Please explain in detail.
1. The methods introduced in this paper are not very relevant to ‘biomolecules’. The author should include some biomolecule analysis methods such as Raman spectroscopy and IR spectroscopy.
Comments on the Quality of English LanguageNA
Reviewer 2 Report
Comments and Suggestions for Authors
The authors consider the state-of-the-art machine and deep learning methods applied for studies aimed at automatic detection of bone related abnormalities in model and aquaculture fish. in general the paper is well written and contains a sufficient amount of information related to the subject matter.
However in my opinion it lacks a section reviewing the recording techniques applied for acquisition of images which are then subjected to processing by AI. It is especially important since the imaging methods are quite different for small model fish and aquaculture ones. I would also suggest some discussion on not very commonly used optical techniques such as OCT and holography.
Reviewer 3 Report
Comments and Suggestions for Authors
A very interesting review manuscript related to a cutting-edge subject, i.e. the application of image analysis methods and Artificial Intelligence (AI) currently applied in the analysis of bone-related deformities in aquaculture and model fish. The subject is very interesting not only for financial reasons (i.e. fish health in aquaculture) but also for environment reasons since such skeletal deformities are indicators of polluting factors.
The manuscript is well organized, written and presented without linguistic issues.
Only two comments:
1. In figure 2, propose to replace “Without phenotype” with “Normal”
2. Propose to add a separate section for the image analysis software tools and AI tools used in the presented bibliography to address the subject matter
Round 2
Reviewer 1 Report
Comments and Suggestions for Authors
The authors addressed all the questions properly.